# Saline–Alkaline Stress Resistance of Cabernet Sauvignon Grapes Grafted on Different Rootstocks and Rootstock Combinations

**DOI:** 10.3390/plants12152881

**Published:** 2023-08-06

**Authors:** Baolong Zhao, Zhiyu Liu, Chunmei Zhu, Zhijun Zhang, Wenchao Shi, Qianjun Lu, Junli Sun

**Affiliations:** 1Department of Horticulture, College of Agriculture, Shihezi University, Shihezi 832003, China; zhaobaolong@stu.edu.cn (B.Z.); liuzhiyu@stu.shzu.edu.cn (Z.L.); zhuchunmei@stu.shzu.edu.cn (C.Z.); zhangzhijun85@xjshzu.com (Z.Z.); shiwenchao@stu.shzu.edu.cn (W.S.); liuzhiyu@xjshzu.com (Q.L.); 2The Key Laboratory of Special Fruits and Vegetables Cultivation Physiology and Germplasm Resources Utilization of the Xinjiang Production and Construction, Shihezi 832003, China

**Keywords:** saline–alkaline stress, grapes, photosynthetic fluorescence, rootstocks, grafting

## Abstract

Grafting the wine grape variety Cabernet Sauvignon onto salinity-tolerant rootstocks can improve salinity tolerance and grape yields in regions with high salinity soils. In this experiment, the effects of different rootstocks and rootstock combinations on the saline–alkaline stress (modified Hoagland nutrient solution + 50 mmol L^−1^ (NaCl + NaHCO_3_)) of Cabernet Sauvignon were studied. Correlation and principal component analyses were conducted on several physiological indicators of saline–alkaline stress. Salinity limited biomass accumulation, induced damage to the plant membrane, reduced the chlorophyll content and photosynthetic capacity of plants, and increased the content of malondialdehyde, sodium (Na^+^)/potassium (K^+^) ratio, and antioxidant enzyme activities (superoxide dismutase, peroxidase, and catalase). Significant differences in several indicators were observed among the experimental groups. The results indicate that the saline–alkaline tolerance of Cabernet Sauvignon after grafting was the same as that of the rootstock, indicating that the increased resistance of Cabernet Sauvignon grapes to saline–alkaline stress stems from the transferability of the saline–alkaline stress resistance of the rootstock to the scion.

## 1. Introduction

The yield of cultivated plants is affected by natural environmental factors, and soil salinization has become an increasingly serious global problem [1]. More than 800 million hectares of arable land are affected by salinity, and approximately 25% of China’s farmland is estimated to be salinized [2]. The growth of most crops and fruits is inhibited by high saline–alkaline stress, and these plants are unable to complete their normal life cycles under high saline–alkaline stress [3]. Soil salinization has thus become a major problem, given that it has become one of the major factors limiting the production of crops worldwide [4]. Soil salinity results in both salt stress and alkaline stress. Saline conditions can be classified as mild (less than 3‰ salt content, pH 7.1–8, and pH over 9.5) according to the salt content and pH [5]. Many studies have shown that saline–alkaline stress can cause severe nutrient ion imbalances, reduce osmoregulation, inhibit the activity of antioxidant systems, impair photosynthetic capacity, and severely inhibit plant growth [6].

Grafting is an ancient and widespread agronomic technique. The phenotypes of commercial varieties of the aboveground parts of plants can be altered using rootstocks [7,8]. Some of the advantages of using rootstocks include increases in crop yield via enhanced resistance to abiotic stress and increases in land use by permitting the growth of crops in specific areas [9,10,11]. Rootstocks can alter scion physiology and the expression of genes via molecular signaling [12]. Therefore, grafting is commonly used in agriculture to confer varieties with favorable traits. For example, the use of resistant rootstocks in pepper can protect scions from drought-induced oxidative stress and enhance yields in the field [13]. Pumpkin has been used as a rootstock to increase putrescine biosynthesis via the up-regulation of spermidine decarboxylase activity, and this increases the cold tolerance of cucumber [14]. Feng et al. found that grafting tomato onto LBP reduces salt sensitivity by increasing the expression of key enzymes and improving photosynthesis in the scion [15]. In grapes, the grafting of good commercial varieties onto resistant rootstocks can improve yield and abiotic and biotic stress tolerance [16,17].

Cabernet Sauvignon is one of the most widely cultivated wine grape varieties in the world, and Xinjiang has the largest planting area of Cabernet Sauvignon in China [18]. However, most of the soils in Xinjiang are saline, and saline soils can inhibit water absorption, reduce photosynthesis, and affect the normal growth and development of plants, resulting in lower crop yields [19]. Reducing soil salinity is key to enhancing the production of Cabernet Sauvignon grapes. Several studies have demonstrated that the use of rootstocks resistant to saline–alkaline stress can enhance grape yield and quality [20]. Migicovsky et al. found that the use of rootstocks can alter the endogenous substances in plants, and this can result in various internal physiological and biochemical changes in grape plants that can enhance their growth as well as their resistance to environmental stress [21].

Desirable traits of rootstocks can be transmitted to the scion, and previous studies have focused on the identification of superior rootstock varieties, as well as characterizing changes in biomass, fruit growth and development, quality, and the resistance of scion varieties grafted onto different rootstocks. However, few studies have examined the relationship between changes in the resistance of scion varieties grafted onto different rootstocks and the stress resistance of rootstock varieties. A saline–alkaline-tolerant variety (1103P) and a saline–alkaline-sensitive variety (3309M) have been identified in previous studies [22]. In this study, the chlorophyll content, photosynthetic rate, enzyme activity, photosynthetic fluorescence, and ion content of these two rootstock varieties and combinations of rootstocks grafted with Cabernet Sauvignon were measured under saline–alkaline stress. Correlation and principal component analyses were conducted on these physiological indicators to clarify the effects of grafting on the saline–alkaline tolerance of both the scion and rootstock. Our findings provide new insights that will help alleviate the deleterious effects of saline–alkaline stress on crops.

## 2. Results

### 2.1. Effect of Saline–Alkaline Stress on New Shoot Growth and the Membrane System of Different Rootstocks and Rootstock Combinations

The growth of plants in all groups was affected by saline–alkaline stress (Figure 1A,B). After 15 d of stress, the new shoot growth was 105.24%, 81.10%, 165.08%, 89.23%, 70.91%, and 105.95% lower in “CS”, “1103P”, “3309M”, “CS/CS”, “CS/1103P”, and “CS/3309M” than in the control, respectively, indicating that the stress-induced damage to different rootstocks and rootstock combinations varied under saline–alkaline stress. The decrease in shoot growth was the lowest in “1103P” and “CS/1103P”, indicating that their saline–alkaline tolerance was highest. The MDA content and relative conductivity of both ungrafted and grafted plants were significantly higher than those of control plants under saline–alkaline stress (Figure 1C,D), and the MDA content and relative conductivity of “1103P” and “CS/1103P” were lower than those of the self-rooted vine plantlets and “3309M” vine plantlets; stress-induced damage to the former plants was also lower than that of the latter plants.

### 2.2. Effect of Saline–Alkaline Stress on the Antioxidant Enzyme Activity of Different Rootstocks and Rootstock Combinations

As can be seen from Figure 2A–C, the antioxidant enzyme activities in different experimental groups did not change substantially during the treatment period, and the superoxide dismutase (SOD), peroxidase (POD), and catalase (CAT) activities increased under saline–alkaline stress. After 15 d of stress, the SOD, POD, and CAT activities of “1103P” were the highest; the activity of SOD, POD, and CAT was 152.11%, 134.22%, and 133.73% higher in “1103P” than in the control, respectively. SOD, POD, and CAT activities were lowest in “3309M”, and they were 83.26%, 51.91%, and 53.42% higher in “3309M” than in the control, respectively. After grafting CS, changes in antioxidant enzyme activities were similar to those observed in the rootstock under saline–alkaline stress; the greatest changes in antioxidant enzyme activities were observed in “CS/1103P”. We also observed no significant changes in the levels of antioxidant enzymes in CS and CS/CS, indicating that grafting alone did not alter the saline–alkaline resistance of Cabernet Sauvignon.

### 2.3. Effect of Saline–Alkaline Stress on the Chlorophyll Content and Photosynthetic Properties of Different Rootstocks and Rootstock Combinations

The content of chlorophyll a (Chla) and chlorophyll b (Chlb) in both ungrafted and grafted plants was significantly lower under saline–alkaline stress than under control conditions (Figure 3A,B). The content of Chla and Chlb in self-rooted vine leaves decreased the least under saline–alkaline stress among all vine plantlets exposed to saline–alkaline stress. Similarly, Pn, Gs, Ci, and Tr of the leaves in all experimental groups were significantly lower under saline–alkaline stress than under control conditions (Figure 3C–F). The reductions in the photosynthetic indexes of the rootstock “1103P” and the rootstock combination “CS/1103P” were less pronounced than those observed for “3309M” and “CS/3309M”. The reductions in these photosynthetic indicators were most pronounced in “3309M”.

### 2.4. Effect of Saline–Alkaline Stress on the Ion Content in the Roots and Leaves of Different Rootstocks and Rootstock Combinations

The Na^+^ content in the roots (Figure 4A–C) and leaves (Figure 4D–F) increased significantly and the K^+^ content decreased significantly in all experimental groups under saline–alkaline stress; this resulted in a significant increase in the Na^+^/K^+^ ratio. The Na^+^ content was higher in the roots than in the leaves of different grape rootstocks, and the Na^+^ content in the roots and leaves of all three grape rootstocks was significantly higher after saline–alkaline stress than in control plants. The Na^+^ content was highest in the leaves of ”3309M”, and the leaf Na+ content was 8.33-fold higher in “3309M” than in the control. The Na^+^ content was lowest in “1103P”, and it was 4.33-fold higher in “3309M” than in the control, indicating that iontophoresis was strong in “1103P”. This likely retained Na^+^ in the root, reducing its influence on the aboveground part. The content of K^+^ was lower in all plants under saline–alkaline stress than under control conditions, and the K^+^ content was highest in “1103P” and lowest in “3309M”. Changes in the Na^+^ and K^+^ content after CS grafting were similar to those in the rootstock, indicating that saline–alkaline stress caused Na^+^ to accumulate in the organs of Cabernet Sauvignon and K^+^ to be exuded or absorbed less. By contrast, the more salinity-tolerant rootstocks could alleviate the effects of saline–alkaline stress in Cabernet Sauvignon by reducing the Na^+^ content and increasing the K^+^ content.

### 2.5. Effect of Saline–Alkaline Stress on the Rapid Chlorophyll Fluorescence Induction Curve of Different Rootstocks and Rootstock Combinations

The OJIP curves were normalized to O–P. After normalization, the fluorescence intensity was positively correlated with the degree of injury. As can be seen in Figure 5A, the fluorescence intensity of all ungrafted rootstocks was significantly higher under saline–alkaline stress than under control conditions. Significant differences were observed in the J–I segment; specifically, the fluorescence intensity of “3309M” was highest, followed by Cabernet Sauvignon and “1103P”. After grafting, changes in the fluorescence intensity of Cabernet Sauvignon with different rootstocks were similar to those before grafting (Figure 5B); after saline–alkaline treatment, there were no significant differences in the O section. Significant changes were observed in the J–I section; the fluorescence intensity was highest in “CS/3309M”, which indicates that it was the most damaged. The fluorescence intensity of “CS/3309M” was significantly lower than that of self-rooted vines and “CS/3309M”. This indicates that saline–alkaline stress reduces the fluorescence effect of different rootstocks and rootstock combinations, and the rootstock “1103P” can alleviate the damage induced by saline–alkaline stress to Cabernet Sauvignon.

### 2.6. Effect of Saline–Alkaline Stress on the Fluorescence Parameters of Different Rootstocks and Rootstock Combinations

The chlorophyll fluorescence parameters of different rootstocks and rootstock combinations under saline–alkaline stress are shown in Figure 6A,B. The performance index based on light energy absorption (PI abs) of the rootstock “1103P” was higher than that of the other varieties; the maximum photochemical efficiencies of PSII (Fv/Fm) of saline–alkaline stress “1103P” and “cs/1103P” were 0.7773 and 0.8097, respectively, which were the highest values in the rootstock and scion–rootstock combinations, compared with the control; and the heat dissipation per unit area, DIo/CSm, was highest in “CS”. After 15 d of saline–alkaline stress, the relative variable fluorescence intensity at the J step (Vj), the relative variable fluorescence intensity at the I step (Vi), and the light energy absorbed per unit RC (ABS/RC) and the light energy captured per unit RC (TRo/RC) were higher for each rootstock than in control plants. The light energy absorbed per unit RC (ABS) and light energy captured per unit RC (TRo/RC) were increased; increases and decreases were observed in DIo/CSm, and decreases were observed in the other fluorescence parameters. The values of ABS/RC, TRo/RC, and DIo/CSm did not differ significantly among rootstocks and rootstock combinations; the smallest decreases in PI abs and Fv/Fm were observed in “1103P”. This indicates that rootstock leaves with high salinity tolerance can improve the photochemical utilization of Cabernet Sauvignon and alleviate the damage caused by saline–alkaline stress to Cabernet Sauvignon leaves by improving the absorption, capture, and transfer of light energy as well as reducing heat dissipation.

### 2.7. Correlation and Principal Component Analysis of Several Indicators of Different Rootstocks and Rootstock Combinations under Saline–Alkaline Stress

The correlation analysis showed that grape growth was significantly positively correlated with Gs, Ci, K^+^, and Vi and negatively correlated with antioxidant enzymes (Figure 7A); all chlorophyll fluorescence indicators were significantly negatively correlated with antioxidant enzymes, with the exception of Vj and ABS/RC, which were positively correlated with antioxidant enzymes. Gs, Ci, Tr, and Pn, which affect the photosynthetic rate, were significantly positively correlated with chlorophyll fluorescence. Na^+^ was significantly positively correlated with MDA, conductivity, and antioxidant enzymes, and K^+^ was negatively correlated with MDA, conductivity, and antioxidant enzymes. Chlorophyll was significantly positively correlated with Gs, Ci, Tr, and Pn, which affect photosynthesis, and significantly negatively correlated with antioxidant enzymes and the content of MDA. Principal component plots based on the first and second principal components were used to analyze changes in the responses of different rootstocks and rootstock combinations to saline–alkaline stress (Figure 7B,C). A total of 26 indicators loaded significantly on the extracted principal components, and the cumulative variance explained by the first two principal components was 80.6% and 82.5% for the different rootstocks and rootstock combinations, respectively. The rootstocks and rootstock combinations were significantly separated from the control under saline–alkaline stress according to the principal component analysis; saline–alkaline stress altered the growth status of grapes, and significant differences were observed among rootstocks and rootstock combinations.

## 3. Discussion

Under saline–alkaline stress, the normal growth and development of plants are severely disrupted [23]. Saline–alkaline stress has been shown to significantly inhibit the growth of vine shoots in a study of “Red Globe” grapes [24]. The height of the vines decreases as the salt concentration increases. After vines are subjected to salt stress, the leaves of the plants become yellow and wilted, and their growth is significantly reduced [25]. In this experiment, vine plantlets were subjected to saline–alkaline stress. The growth of vines was inhibited following exposure to saline–alkaline stress (Figure 1), but differences were observed among the different rootstocks and rootstock combinations; the reductions in the new shoot growth of “1103P” and “CS/1103P” were similar. The degree of growth inhibition depended on the relative saline–alkaline stress tolerance of plants; the saline–alkaline stress tolerance of “1103P” was high, and this is consistent with the results of previous studies [26].

Saline–alkaline stress-induced membrane lipid peroxidation leads to the production and accumulation of reactive oxygen species (ROS), and the content and relative conductivity of MDA reflect the extent of damage to plant cell membranes. Luo et al. [27] explored the mechanism of salinity-induced ROS production and found that the overexpression of genes encoding ABI4 transcription factors under salt stress promotes the accumulation of ROS, which disrupts the normal physiological functions of plants. The scavenging system in plants reduces the deleterious effects of ROS stress on plants. In 10 table grapes, Cynthia Lu et al. [28] found that the MDA and H_2_O_2_ content in grape leaves increased gradually as the saline–alkaline concentration increased; this stimulated the antioxidant defense system of plants and significantly increased the activity of antioxidant enzymes. Saline–alkaline treatment of Kyoho grapes was shown to down-regulate the relative expression of the antioxidant enzyme genes *VvSOD*, *VvPOD*, *VvCAT*, and *VvAPX*; decrease the activities of SOD, POD, CAT, and ascorbate peroxidase in vine shoots and leaves; and result in significant increases in the leaf superoxide anion (O^2-^), hydrogen peroxide (H_2_O_2_), and malondialdehyde (MDA) content as well as relative conductivity in grape vines [29]. Yang et al. reported that the enhanced salt tolerance in rootstock-grafted plants was associated with a significant increase in the activity of scavengers in the antioxidant system (SOD, POD, and CAT) in the rootstock [30]. In this experiment, SOD, POD, and CAT activities were significantly higher in the leaves of all test groups under saline–alkaline stress than under control conditions. However, rootstocks with high salinity tolerance could resist the deleterious effects of oxidative stress by enhancing ROS scavenging, and “1103P” and “CS/1103P” produced more antioxidant enzymes. Damage to these two plants was lower than the damage to plants observed in groups with other rootstock combinations.

Chlorophyll can directly affect the photosynthetic efficiency of plants, and the content of chlorophyll provides a measure of the ability of the stress tolerance of plants [31]. Previous studies have shown that the chloroplasts of plants are damaged by saline–alkaline stress, and this leads to the degradation of chloroplast pigments, which results in an imbalance in the photosynthetic system and affects normal plant growth [32]. In our study, the leaves of grapes became yellow, and the Chla and Chlb content of the plants significantly decreased under saline–alkaline stress; this probably stems from the disruption of intracellular ion homeostasis of plants, which results in the inhibition of chlorophyll synthesis due to decreases in the Mg^2+^ content [33]. The content of Chla and Chlb was significantly reduced in all experimental groups. Decreases in the chlorophyll content of “1103P” and “CS/1103P” under saline–alkaline stress were the lowest among all vine plantlets, indicating that “1103P” rootstocks were highly resistant to saline–alkaline stress; this finding is consistent with the results of a previous study.

Saline–alkaline stress can affect photosynthesis by mediating various physiological changes in plants, and changes in plant photosynthesis caused by stomatal closure are considered an early response to saline–alkaline stress [34]. Treatment of Amur grapes (*V. amurensis* Rupr.) with different levels of saline–alkaline stress revealed that their photosynthetic fluorescence systems were all damaged to varying degrees, and Pn, Gs, Tr, and Ci decreased [35]. In our study, Pn, Gs, Ci, and Tr were significantly lower in leaves than in vine plantlets under saline–alkaline stress than under control conditions after 15 d of treatment; the lowest reductions in photosynthetic indicators were observed in “1103P” and “CS/1103P”. This indicates that “1103P” can reduce the inhibition of photosynthesis in Cabernet Sauvignon plants caused by saline–alkaline stress [36].

Saline–alkaline stress disrupts ion homeostasis in plants, and excessive levels of Na^+^ in plants impede the uptake of other ions; the concentrations of these ions affect the chlorophyll content and photoelectron transfer, which affect photosynthesis [37]. Gupta et al. found that the wilting of plant leaves stems from the presence of excess Na^+^ in the roots and leaves, which prevents water uptake, leads to water deficiency and osmotic stress, and affects normal plant growth [38]. In general, the K^+^ concentration is closely related to plant osmosis, the regulation of membrane potential, and enzyme activity [39]; a decrease in the K^+^ concentration has a negative effect on the plant antioxidant system, and a lower Na^+^/K^+^ ratio in the cytoplasm is also necessary for normal plant growth and development. Previous studies have suggested that non-saline plants such as grapes can rely on the ability of the root system to limit Na^+^ uptake to maintain low Na^+^ levels in the leaves [40]. Subsequent studies in pumpkin revealed that genes such as HKT1, SOS1, and RBOHD/F are involved in regulating the transport and distribution of Na^+^ and K^+^ in rootstocks, which in turn regulate salt tolerance in grafted plants [41,42,43]. In tomato, HKT1:1 and HKT1:2 are involved in regulating the salt tolerance of grafted plants [44]. These findings are consistent with the results of our study. Na^+^ levels increased and K^+^ levels decreased in plants under saline–alkaline stress; Na^+^ levels were higher and K^+^ levels were lower in the root system than in the leaves. By contrast, decreases in the K^+^ content affect the activity of related enzymes induced by K^+^ [45]. Therefore, the more salinity-tolerant plant species generally have a greater ability to absorb and transport K^+^ upwards, as well as lower Na^+^/K^+^ values [46]. In this study, the increases in the Na+ content and decreases in the K^+^ content were lower for “1103P” and “CS/1103P” under saline–alkaline stress compared with Cabernet Sauvignon self-rooted vines and “3309M”, indicating that the ion-blocking properties of “1103P” rootstocks were stronger than those of other plants.

The OJIP curve provides a visual reflection of the extent of stress-induced plant damage [47]. In this experiment, the fluorescence intensity of the O and J phases increased, the fluorescence intensity of the I and P phases decreased in the different rootstocks and rootstock combinations under salinity stress, and the increased fluorescence intensity of the O phase indicated that the deleterious effects of stress were reversible; the increased fluorescence intensity of the J phase indicated that the PSII receptor was damaged and the electron transfer from QA to QB was restricted [48]. The O and J phase fluorescence intensity of treated grape plants for different rootstock and rootstock combinations was lower than that of Cabernet Sauvignon self-rooted vines, and the lowest increases in the O and J phase fluorescence intensity were observed for “CS/1103P” and “1103P”, which were the least affected by saline–alkaline stress. This indicates that “1103P” can accelerate the electron transfer from QA to QB more than “3309M” and increase the fluorescence intensity of the plant after they are grafted on Cabernet Sauvignon, which alleviates the deleterious effects of saline–alkaline stress.

Chlorophyll fluorescence parameters reflect a series of regulatory processes within the plant photosynthetic apparatus and indicate the extent to which the photosynthetic apparatus is disrupted in terms of electron transfer and energy conversion; they are thus critically important for studies of plant photosynthesis [49]. PI abs and Fv/Fm reflect the degree of photoinhibition in leaves [50]. Amorim et al. found that resistant rootstocks had a beneficial effect on the growth and photosynthesis of grafted grape plants under salinity [51]. In our experiment, the PI abs and Fv/Fm were lower in plants with different rootstocks and rootstock combinations than in the control, indicating that saline–alkaline stress damaged the cellular and photosynthetic structure of the leaves [52]. The decrease in PI abs and Fv/Fm of the leaves was lowest in “CS/1103P” and “1103P”; other fluorescence parameters were also modulated in “CS/1103P” and “1103P”, which resulted in decreases in ABS/CSm, TRo/CSm, ETo/CSm, and RC/CSm. Decreases in ABS/CSm, TRo/CSm, ETo/CSm, RC/CSm, and ETo/RC values and increases in Vj, Vi, ABS/RC, TRo/RC, and DIo/CSm values were low in “CS/1103P” and “1103P” under saline–alkaline stress. This indicates that the “1103P” rootstock can maintain the activity of the photosynthetic center of the plant under saline–alkaline stress, which enhances the ability of the leaves to absorb and capture light energy, increases the flow of electrons through the PSII reaction center, reduces the energy used for reduction and heat dissipation, and regulates the content and ratio of ABS/RC to optimize the distribution of excitation energy. This enhances the structure of the photosynthetic membrane and alleviates the deleterious effects of saline–alkaline stress on grafted vine plantlets [53].

The complexity of saline–alkaline tolerance mechanisms in plants necessitates measurements of a diverse set of indicators. In this study, correlation and principal component analyses were carried out on 26 physiological indicators of saline–alkaline stress. Saline–alkaline stress affected different rootstocks by disrupting intracellular ion homeostasis and the cell membrane system, and the deleterious effects of saline–alkaline stress on plants were mainly alleviated through organic and inorganic osmoregulation and increases in SOD and CAT activities. The stress-alleviating effects of rootstocks were transferred to the scion through grafting, and this made the scion resistant to stress. There is an exchange of long-distance signals and mobile substances in the grafting complex, and mRNA, miRNA, peptides, and functional proteins, in addition to common hormones, mineral elements, and sugars, can be transmitted between rootstock and scion, which in turn affects the traits of grafted plants [54,55]. In our experiment, “1103P” rootstocks had greater salinity tolerance than “3309M” rootstocks, and we confirmed that grafting rootstocks with high salinity resistance increases the salinity resistance of the scion.

## 4. Materials and Methods

The hardwood cuttings of “1103P” (Riparia ×Rupestris hybrid), “3309M” (5BB × SO4 hybrid), and Cabernet Sauvignon were placed in polythene bags, sealed, and housed in sand in November 2020. The sand-housed hardwood cuttings were removed in January 2021 and then cut into 10–15 cm (containing one bud) hardwood cuttings; they were then rooted and germinated in the greenhouse, and cuttings were placed in a nutrient bowl (8 cm × 8 cm × 10 cm) 20 d later, which were filled with the seedling substrate (peat:vermiculite = 2:1, *v*:*v*). The vine plantlets were then moved to a greenhouse and subjected to standard management conditions. The vine plantlets were grafted with “CS” as the scion and “3309M” and “1103P” as the rootstocks by cleft grafting when they reached 8 to 11 leaves: At the same time, “1103P”, “3309M”, and Cabernet Sauvignon vines were subjected to hydroponic culture in pots. The treatments were initiated after 10 d of slowed growth; in the control, only nutrient solution was applied; and in saline–alkaline stress treatments, nutrient solution and 50 mmol L^−1^ (NaCl + NaHCO_3_) (mixed in a 1:1 mass ratio, pH = 8.3) were applied. There were three replicates for each treatment, and the nutrient solution was changed every 5 days. There were a total of three treatments.

The growth of new shoots was measured 0 and 15 days after treatment, and the length of the aerial part of the plant was measured using a tape measure 0 and 15 days after treatment, and the average value was taken to be 0.1 cm. Relative conductivity was measured using a conductivity meter 15 d after treatment. Three 0.1 g samples were placed in a graduated test tube with 10 mL of deionized water, covered with a glass stopper, and left to soak at room temperature for 12 h. The conductivity of the extract (R1) was determined using a conductivity meter; it was then heated in a boiling water bath for 30 min, cooled to room temperature, shaken well, and the conductivity of the extract (R2) was determined again. Relative conductivity = R1/R2 × 100%. The malondialdehyde (MDA) content was measured using thiobarbituric acid 15 d after treatment.

Samples were taken 15 d after treatment to determine peroxidase (POD) activity using the guaiacol method; superoxide dismutase (SOD) activity was measured using the nitrogen blue tetrazolium method; and catalase (CAT) activity was measured using the guaiacol method [56].

Fresh leaves were collected 15 d after treatment; they were then wiped clean and cut with the veins removed. Next, 0.2 g of leaf tissue was placed in a test tube and added to 10 mL of 95% ethanol; chlorophyll was extracted until the leaves turned white, and the leaves were protected from light. The absorbance values at 470, 649, and 665 nm were measured using a spectrophotometer, and Arnon’s formula was used to calculate the chlorophyll content (Arnon, 1949). Net photosynthetic rate (Pn), stomatal conductance (Gs), transpiration rate (Tr), and sub-stomatal CO_2_ concentration (Ci) of the functional leaves of each rootstock species were measured 15 d after treatment (9:00–11:00 a.m.) using a Li-6400XT portable photosynthesizer (LI-COR, Redmond, WA, USA) equipped with red and blue light sources (light quantum flux density of 1000 μmol m^–2^ s^–1^).

Fresh leaf and root samples (0.1 g) were placed in reaction jars, and 8 mL of nitric acid + 2 mL of hydrochloric acid were added. After the mixture was shaken well and left to stand overnight, an airtight microwave digestion apparatus (CEM MARS6, Changyi Scientific Instrument Co., LTD, Shanghai, China) was used to heat the digestion, and the mixture was digested until the liquid was colorless or clear. After acid was applied at 150 °C, the mixture was cooled to room temperature, and the liquid was transferred to a 50 mL volumetric flask. After the volume of the mixture stabilized, it was left to stand for 1 h, and it was transferred to the supernatant in a 5 mL centrifuge tube. Each mineral element was determined using a plasma emission spectrometer (ICAP 6300, Leiden Scientific Instruments LTD, Suzhou, China).

Rapid chlorophyll fluorescence induction kinetic curves (OJIP curves) of leaves were measured at 15 d after treatment using a Multifunctional Plant Efficiency Analyzer (Hansatech Instruments Limited, Norfolk, UK). Measurements were conducted over 2 s (red light induction), and leaves were exposed to 30 min of dark treatment prior to measurements. The O, J, I, and P on the OJIP curve corresponded to 0.00002 s, 0.002 s, 0.03 s, and 0.4–1 s, respectively. JIP parameters were calculated following the method of Hu Wenhai [57]. Relevant parameters and their meanings are shown in Appendix A.

Microsoft Excel 2015 (Microsoft, Redmond, WA, USA) was used for data processing and analysis. IBM SPSS Statistics 25 (IBM Corporation, Armonk, NY, USA) was used for data analysis. Pearson correlation coefficients were used to conduct correlation analyses. The threshold for statistical significance was *p* < 0.05. The data were plotted using Origin 2021 (OriginLab Corporation, MA, USA) and Sigmaplot 12.5 (Systat Software, Inc., Richmond, CA, USA) software.

## 5. Conclusions

In summary, the normal growth and development of grapes are severely disrupted under saline–alkaline stress. The morphological, physiological, and molecular responses of different rootstocks, which differ in their abilities to maintain ionic homeostasis, regulate osmotic stress, and scavenge ROS, can be transferred to the scion through grafting. We found that salinity tolerance in grafted Cabernet Sauvignon was identical to that of the rootstock and that the enhanced resistance of Cabernet Sauvignon grapes under saline–alkaline stress stemmed from the transferability of the saline–alkaline stress tolerance from the rootstock to the scion. Overall, the results of our study will help alleviate the deleterious effects of salinity on crops.

## Figures and Tables

**Figure 1 plants-12-02881-f001:**
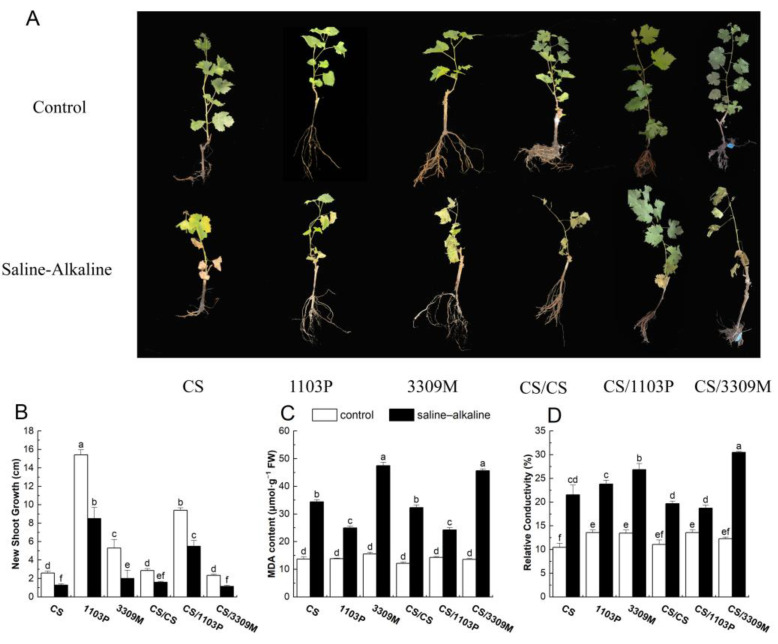
Comparisons of the (**A**) grape phenotype, (**B**) new shoot growth, (**C**) MDA content, and (**D**) relative conductivity of different rootstock and rootstock combinations under saline–alkaline stress. Values were shown as means ± standard deviations (SD, n = 3). Different lowercase letters indicate statistically significant differences between the four treatments at the 0.05 level. Scale bar, 5 cm.

**Figure 2 plants-12-02881-f002:**
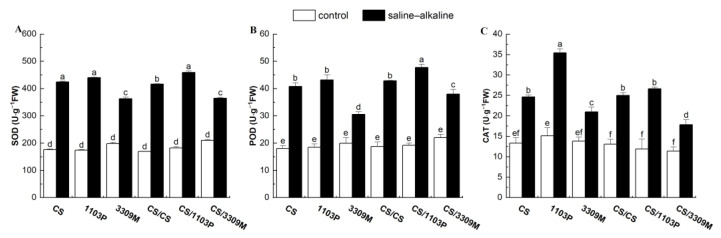
Comparisons of the (**A**) SOD content, (**B**) POD content and (**C**) CAT content of different rootstock and rootstock combinations under saline–alkaline stress. Values were means ± standard deviations (SD, n = 3). Different lowercase letters indicate statistically significant differences between the four treatments at the 0.05 level.

**Figure 3 plants-12-02881-f003:**
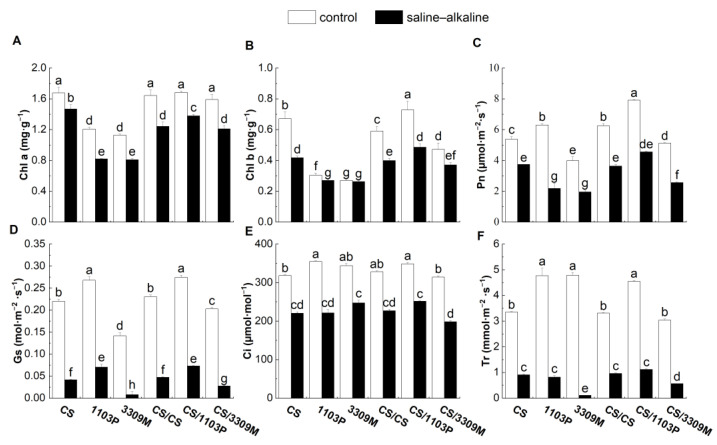
Comparisons of the (**A**) Chl a content, (**B**) Chl b content, (**C**) net photosynthetic rate (Pn), (**D**) stomatal conductance (Gs), (**E**) sub-stomatal CO^2^ concentration (Ci) and (**F**) transpiration rate (Tr) of different rootstock and rootstock combinations under saline–alkaline stress. Values were shown as means ± standard deviations (SD, n = 3). Different lowercase letters indicate statistically significant differences between the four treatments at the 0.05 level.

**Figure 4 plants-12-02881-f004:**
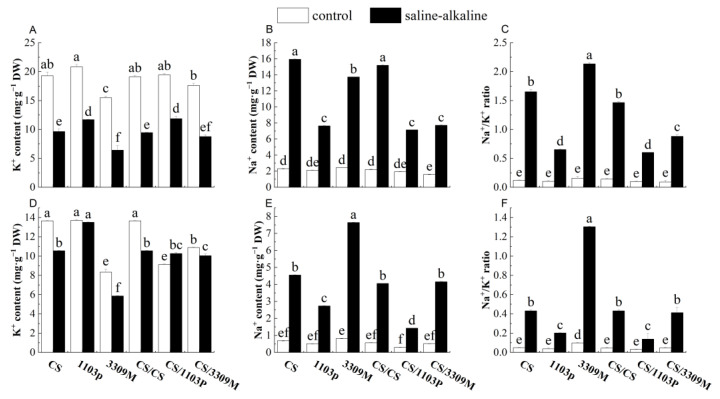
Comparisons of (**A**) K^+^ content, (**B**) Na^+^ content and, (**C**) Na^+^/K^+^ ratio of different rootstock and rootstock combinations in roots under saline–alkaline stress. Comparisons of the (**D**) K^+^ content, (**E**) Na^+^ content and (**F**) Na^+^/K^+^ ratio of different rootstock and rootstock combinations in leaves under saline–alkaline stress. Values were shown as means ± standard deviations (SD, n = 3). Different lowercase letters indicate statistically significant differences between the four treatments at the 0.05 level.

**Figure 5 plants-12-02881-f005:**
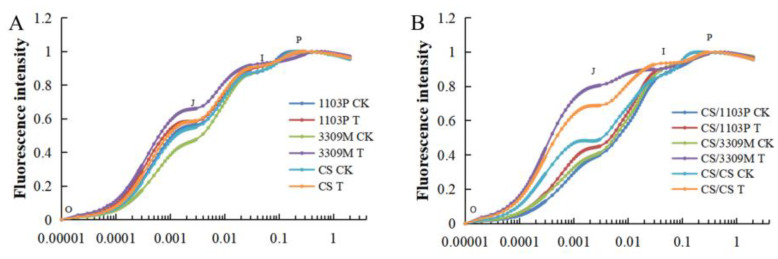
Comparison of the chlorophyll fluorescence induction curves of (**A**) different rootstocks and (**B**) different rootstocks combinations under saline–alkaline stress. The O, J, I, and P on the OJIP curve corresponded to 0.00002 s, 0.002 s, 0.03 s, and 0.4–1 s, respectively.

**Figure 6 plants-12-02881-f006:**
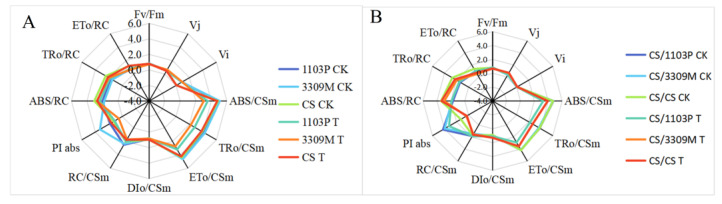
Comparison of the fluorescence parameters of (**A**) different rootstocks and (**B**) different rootstock combinations under saline–alkaline stress. Note: Fv/Fm: Maximum photochemical efficiency of PSⅡ, Vj: Fluorescence intensity at the J step, Vi: Fluorescence intensity at the I step, ABS/CSm: Light energy absorbed per unit area, TRo/CSm: Light energy captured per unit area, ETo/CSm: Quantum yield of electron transport per unit area, DIo/CSm: Heat dissipation per unit area, RC/CSm: The number of reaction centers per unit area, ABS/RC: Light energy absorbed per unit RC, TRo/RC:Light energy captured per unit RC, ETo/RC: Quantum yield of electron transport per unit R, PIabs: Performance index based on light energy absorption.

**Figure 7 plants-12-02881-f007:**
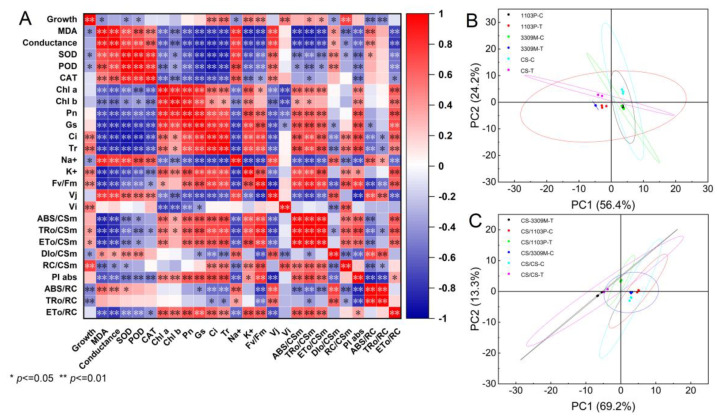
(**A**)Correlation analysis of the physiological indexes of different rootstocks and rootstock combinations, (**B**) Principal component analysis of different rootstocks and (**C**) principal component analysis of different rootstock combinations under saline–alkaline stress. Significance levels of one-way ANOVA: *, 0.01 < *p* < 0.05, significant; **, *p* < 0.01, highly significant; ns, *p* ≥ 0.05, not significant.

## Data Availability

Not applicable.

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
