# Peer review of "Saline–Alkaline Stress Resistance of Cabernet Sauvignon Grapes Grafted on Different Rootstocks and Rootstock Combinations"

_plants, 2023, doi:10.3390/plants12152881_

Round 1
Reviewer 1 Report
The manuscript submitted by Zhao et al. describes the improvement of stress tolerance of the wine grape variety Cabernet Sauvignon by grafting onto the salt-tolerant rootstock. The authors analyzed not only the scion phenotype, but also physiological and molecular responses. The topics described are interesting. However, I have found some issues as follows that need to be addressed and corrected.
Most importantly, what are the signals and molecular mechanisms that activate stress tolerance in the scion? These should be discussed.
Figure 1A. Scale bar should be added.
Line 410; There are no data to support a morphological response under stress conditions, although growth may have been affected.
Minor comments
Line 103 and others; Figure 2 shows the results for SOD, POD, and CAT. Therefore, they should be mentioned in the same order in the text.
Lines 234 and 235; ‘saline’ should be “Saline’.
Line 409; sum should be summary.
Abbreviations such as SOD, POD and CAT should be listed with their full names.
For my opinion, the quality of English on this paper is acceptable, the text is partly fluent but not always.
Author Response
We feel great thanks for your professional review work on our article. As you are concerned, there are several problems that need to be addressed. According to your nice suggestions, we have made extensive corrections to our previous draft, the detailed corrections are listed below.

Reviewer 2 Report
The authors Zhao et al., present an article about Cabinet Sauvignon, saline-alkaline stress, and rootstocks. The authors do not clearly distinguish between previous work on grapes and on cabinet sauvignon in particular. That is what is the knowledge gap that they are addressing? The text seems unnecessarily long for the amount of information presented.

Author Response

(The authors gave the same response as above.)

Reviewer 3 Report
In the submitted manuscript, Zhao et al. compared the effects of different rootstocks and rootstock combinations on several physiological indicators of Cabernet Sauvignon in response to saline-alkaline stress, and concluded that the increased resistance of Cabernet Sauvignon grapes to saline-alkaline stress depends on the grafted rootstock, which provides an important theoretical basis for improving the saline-alkaline stress resistance of grapes. Here are my detailed comments.
1. Figure 1B, y-axis, 'New slight growth' should change to 'New shoot growth'. And the author should provide detailed statistical methods in all the figure legends.
2. The resolution of all the images needs to be improved, especially the X-axis.
3. In this draft, most draw my concern was that the effects of different rootstocks and rootstock combinations on some physiological indicators were also different under normal condition. The authors should redesign the assay or change the way of the data presentation by recalculating the relative changes of these physiological indicators in response to saline-alkaline stress compared with the normal condition.
4. Figures 4 and 5 can be merged into one image for the same type of detection.
5. The description in Figure 6 is too simple, and the author needs to provide detailed information. J. What do I and P represent? The same issue for Figure 7.
The Quality of the English Language is fine.
Author Response

(The authors gave the same response as above.)

Reviewer 4 Report
The manuscript reports the effect of 50 mM (NaCl + NaHCO3 ) on Cabernet Sauvignon vine grafted onto two different rootstocks.
First, I have serious objections to the plant material used (shoot growth from 1 to about 15.5 cm 0. Until now, in similar experimentations on grapevine, propagation material more compatible with growing conditions in field has been used.
The introduction section seems to contain little information focusing on the grapevine. -In the Ln 51-52 the authors report: In grapevine the grafting of good commercial varieties of Lycopersicon has been used to reduce salt sensitivity. Wat is the relation between grapevine and Lycopersicon?
In Material and Methods section, the procedures of the experimentation are not specified in detail : The hardwood cuttings used in the experimentation are not specified (length and the number of nodes). When transplanting the plantlets into the pots, what was the size of the pots and what was the plant growth medium. How was the green grafting realized ? (by shield or cleft grafting). The graft used and the grafting success are not specified. How were the rootstocks selected for the experimentation ?. The 3309 M rootstock is unknown. Do the authors mean 3309 Couderc (Riparia x Rupestris hybrid)? The process of relative conductivity measurements may be explained in detail.
In the results section:
-How many days after the salinity treatment, the first toxicity symptoms appeared in leaves? (visible senescence and marginal leaf necrosis). Can the authors provide photos with leaf symptoms?
- Values of PSII maximum quantum yield (Fv/Fm) measurements are nor presented in the results.
Moreover, the International viticultural terminology is not used correctly. For example, Seedling is what results from the germination of a seed and not from a cutting or instead of branches may by used the term wood or canes.
In Discussion section there is not sufficient analyse in relation to previous works on grapevine (only three references concern the grapevine).
So the paper should be rewritten considering the reviewers’ comments and English language and concepts should be improved.
English language and concepts should be improved.
Author Response
Thank you for your decision and constructive comments on my manuscript. We have carefully considered the reviewers' suggestions and made some changes. We tried our best to improve and made some revisions to the manuscript.

Round 2
Reviewer 4 Report
In my previous revision I noted some mistakes in the terminology related to viticulture. However, they were not considered by the authors. For example, in several points in the text the term grape canes are used instead of the term vine shoots.
Some Values of PSII maximum quantum yield (Fv/Fm) measurements in the text should be provided to compare with previous works (the differences in figure 6 are not clearly visible).
The manuscript contains many other flaws relating to the English language or the corresponding references:
In the Ln 60-61 the authors report:
Several studies have demonstrated that the use of saline-alkaline resistant rootstocks can enhance grape yield and quality [20] . However, the reference [20] concerns eight apple rootstock genotype.
In the Ln 398 the authors report: The branches of 1103 P (Riparia X Rupestris hybrid),3309M ….. instead of Hardwood cuttings of …..
in my opinion the paper should be rewritten or revised by a native English speaking person.
Author Response
Thank you very much for your patient response and the reviewers' scientific and rigorous approach to reviewing the manuscript. Here is the response to the reviewer's comments.

Round 3
Reviewer 4 Report
All my comments in the last revision were not taken into account. For example in the Ln 398 : not branches but hardwood cuttings. In the Lns,127, 261, 404 and 407 : not vine shoots but vines or vine plantlets. In the Ln 126 : not vine shoots but vine leaves .
Moreover, in some places the English language and concepts need to be improved. For example in Lns 151-152 : This likely trapped Na + in the root and reducing .....
In some places the English language and concepts need to be improved. For example in Lns 151-152 : This likely trapped Na + in the root and reducing .....
Author Response
感谢您的来信和审稿人对我们题为的手稿的评论。这些意见对我们的论文的修改和完善都很有价值,对我们的研究具有重要的指导意义。我们仔细研究了评论并进行了更正。修订部分在纸张中以红色标记。
